# LAVIE: HIGH-QUALITY VIDEO GENERATION WITH CASCADED LATENT DIFFUSION MODELS

## ABSTRACT

This work aims to learn a high-quality text-to-video (T2V) generative model by leveraging a pre-trained text-to-image (T2I) model as a basis. It is a highly desirable yet challenging task to simultaneously **a)** accomplish the synthesis of visually realistic and temporally coherent videos while **b)** preserving the strong creative generation nature of the pre-trained T2I model. To this end, we propose **LaVie**, an integrated video generation framework that operates on cascaded video latent diffusion models, comprising a base T2V model, a temporal interpolation model, and a video super-resolution model. Our key insights are two-fold: **1)** We reveal that the incorporation of simple temporal self-attentions, coupled with rotary positional encoding, adequately captures the temporal correlations inherent in video data. **2)** Additionally, we validate that the process of joint image-video fine-tuning plays a pivotal role in producing high-quality and creative outcomes. To enhance the performance of LaVie, we contribute a comprehensive and diverse video dataset named **Vimeo25M**, consisting of 25 million text-video pairs that prioritize quality, diversity, and aesthetic appeal. Extensive experiments demonstrate that LaVie achieves state-of-the-art performance both quantitatively and qualitatively. Furthermore, we showcase the versatility of pre-trained LaVie models in various long video generation and personalized video synthesis applications.

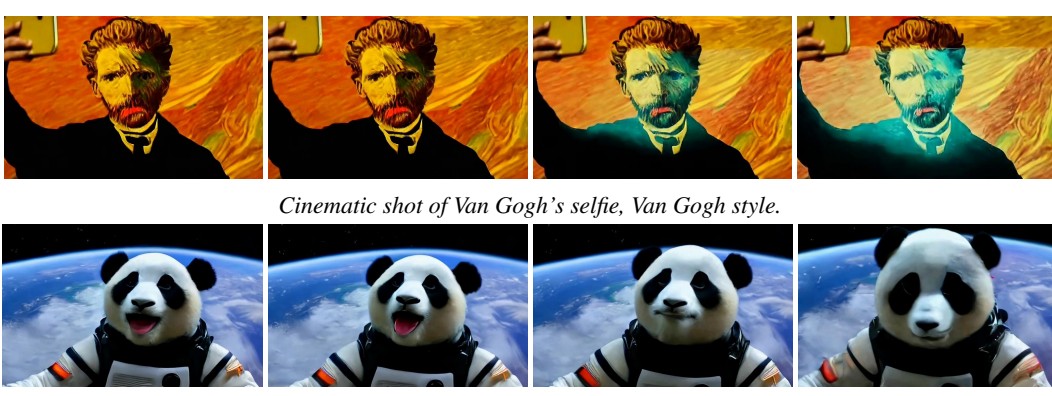

*Cinematic shot of Van Gogh's selfie, Van Gogh style.*

*A happy panda walking in the space.*

Figure 1: **Text-to-video samples.** LaVie is able to synthesize diverse, creative, high-definition videos with photorealistic and temporal coherent content by giving text descriptions.

## 1 INTRODUCTION

With the remarkable breakthroughs achieved by Diffusion Models (DMs) (Ho et al., 2020; Song et al., 2021a;b) in image synthesis, the generation of photorealistic images from text descriptions (T2I) (Ramesh et al., 2021; 2022; Saharia et al., 2022; Balaji et al., 2022; Rombach et al., 2022) has taken center stage, finding applications in various image processing domain such as image out-painting (Ramesh et al., 2022), editing (Zhang & Agrawala, 2023; Mokady et al., 2022; Parmar et al., 2023; Huang et al., 2023; Jiang et al., 2021b) and enhancement (Saharia et al.; Wang et al., 2023a). Building upon the successes of T2I models, there has been a growing interest in extending these techniques to the synthesis of videos controlled by text inputs (T2V) (Singer et al., 2023; Ho

et al., 2022a; Blattmann et al., 2023; Zhou et al., 2022a; He et al., 2022), driven by their potential applications in domains such as filmmaking, video games, and artistic creation.

However, training an entire T2V system from scratch (Ho et al., 2022a) poses significant challenges as it requires extensive computational resources to optimize the entire network for learning spatio-temporal joint distribution. An alternative approach (Singer et al., 2023; Blattmann et al., 2023; Zhou et al., 2022a; He et al., 2022) leverages the prior spatial knowledge from pre-trained T2I models for faster convergence to adapt video data, which aims to expedite the training process and efficiently achieve high-quality results. However, in practice, finding the right balance among video quality, training cost, and model compositionality still remains challenging as it requires careful design of model architecture, training strategies and the collection of high-quality text-video datasets.

To this end, we introduce **LaVie**, an integrated video generation framework (with a total number of 3B parameters) that operates on cascaded video latent diffusion models. LaVie is a text-to-video foundation model built based on a pre-trained T2I model (*i.e.* Stable Diffusion (Rombach et al., 2022)), aiming to synthesize visually realistic and temporally coherent videos while preserving the strong creative generation nature of the pre-trained T2I model. Our key insights are two-fold: 1) simple temporal self-attention coupled with RoPE (Su et al., 2021) adequately captures temporal correlations inherent in video data. More complex architectural design only results in marginal visual improvements to the generated outcomes. 2) Joint image-video fine-tuning plays a key role in producing high-quality and creative results. Directly fine-tuning on video dataset severely hampers the concept-mixing ability of the model, leading to *catastrophic forgetting* and the gradual vanishing of learned prior knowledge. Moreover, joint image-video fine-tuning facilitates large-scale knowledge transferring from images to videos, encompassing scenes, styles, and characters. In addition, we found that current publicly available text-video dataset WebVid10M (Bain et al., 2021), is insufficient to support T2V task due to its low resolution and watermark-centered videos. Therefore, to enhance the performance of LaVie, we introduce a novel text-video dataset **Vimeo25M** which consists of 25 million high-resolution videos ($> 720p$) with text descriptions. Our experiments demonstrate that training on Vimeo25M substantially boosts the performance of LaVie and empowers it to produce superior results in terms of quality, diversity, and aesthetic appeal (see Fig. 1).

## 2 PRELIMINARY OF DIFFUSION MODELS

**Diffusion models (DMs)** (Ho et al., 2020; Song et al., 2021a;b) aim to learn the underlying data distribution through a combination of two fundamental processes: *diffusion* and *denoising*. Given an input data sample $z \sim p(z)$, the diffusion process introduces random noises to construct a noisy sample $z_t = \alpha_t z + \sigma_t \epsilon$, where $\epsilon \sim \mathcal{N}(0, \mathrm{I})$. This process is achieved by a Markov chain with T steps, and the noise scheduler is parametrized by the diffusion-time $t$, characterized by $\alpha_t$ and $\sigma_t$. Notably, the logarithmic signal-to-noise ratio $\lambda_t = log[\alpha^2 t/\sigma^2 t]$ monotonically decreases over time. In the subsequent denoising stage, $\epsilon$-prediction and $v$-prediction are employed to learn a denoiser function $\epsilon_\theta$, which is trained to minimize the mean square error loss by taking the diffused sample $z_t$ as input:

$$\mathbb{E}_{\mathbf{z} \sim p(z), \ \epsilon \sim \mathcal{N}(0,1), \ t} \left[ \| \epsilon - \epsilon_\theta(\mathbf{z}_t, t) \|_2^2 \right]. \tag{1}$$

**Latent diffusion models (LDMs)** (Rombach et al., 2022) utilize a variational autoencoder architecture, wherein the encoder $\mathcal{E}$ is employed to compress the input data into low-dimensional latent codes $\mathcal{E}(z)$. Diverging from previous methods, LDMs conduct the diffusion and denoising processes in the latent space rather than the data space, resulting in substantial reductions in both training and inference time. Following the denoising stage, the final output is decoded as $\mathcal{D}(z_0)$, representing the reconstructed data. The objective of LDMs can be formulated as follows:

$$\mathbb{E}_{\mathbf{z} \sim p(z), \ \epsilon \sim \mathcal{N}(0,1), \ t} \left[ \| \epsilon - \epsilon_\theta(\mathcal{E}(\mathbf{z}_t), t) \|_2^2 \right]. \tag{2}$$

Our proposed LaVie follows the idea of LDMs to encode each video frames into per frame latent code $\mathcal{E}(z)$. The diffusion process is operated in the latent spatio-temporal distribution space to model latent video distribution.

## 3 OUR APPROACH

Our proposed framework, LaVie, is a cascaded framework consisting of Video Latent Diffusion Models (V-LDMs) conditioned on text descriptions. The overall architecture of LaVie is depicted in

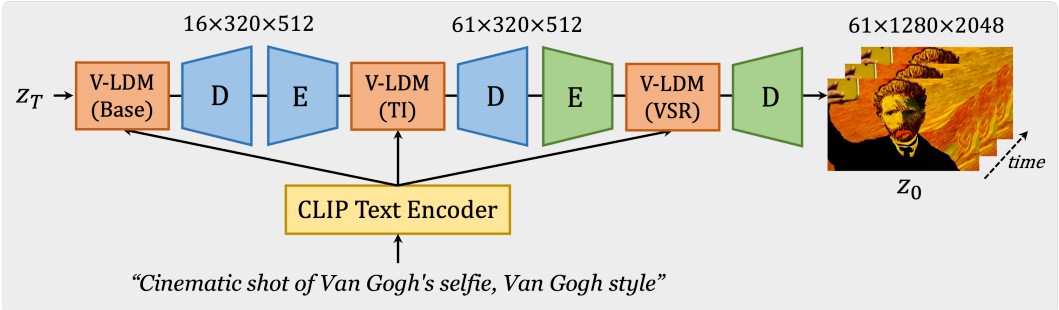

Figure 2: **General pipeline.** LaVie consists of three modules: a Base T2V model, a Temporal Interpolation (TI) model, and a Video Super Resolution (VSR) model. It also requires Encoder (E) and Decoder (D) from pretrained VAE. At the inference stage, given a sequence of noise and a text description, the base model aims to generate key frames aligning with the prompt and containing temporal correlation. The temporal interpolation model focuses on producing smoother results and synthesizing richer temporal details. The video super-resolution model enhances the visual quality as well as elevates the spatial resolution even further. Finally, we generate videos at $1280 \times 2048$ resolution with 61 frames.

Fig. 2, and it comprises three distinct networks: a Base T2V model responsible for generating short, low-resolution key frames, a Temporal Interpolation (TI) model designed to interpolate the short videos and increase the frame rate, and a Video Super Resolution (VSR) model aimed at synthesizing high-definition results from the low-resolution videos. Each of these models is individually trained with text inputs serving as conditioning information. During the inference stage, given a sequence of latent noises and a textual prompt, LaVie is capable of generating a video consisting of 61 frames with a spatial resolution of $1280 \times 2048$ pixels, utilizing the entire system. In the subsequent sections, we will elaborate on the learning methodology employed in LaVie, as well as the architectural design of the models involved.

### 3.1 BASE T2V MODEL

Given the video dataset $p_\text{video}$ and the image dataset $p_\text{image}$, we have a T-frame video denoted as $v \in \mathbb{R}^{T \times 3 \times H \times W}$, where $v$ follows the distribution $p_\text{video}$. Similarly, we have an image denoted as $x \in \mathbb{R}^{3 \times H \times W}$, where $x$ follows the distribution $p_\text{image}$. As the original LDM is designed as a 2D UNet and can only process image data, we introduce two modifications to model the spatio-temporal distribution. Firstly, for each 2D convolutional layer, we inflate the pre-trained kernel to incorporate an additional temporal dimension, resulting in a pseudo-3D convolutional layer. This inflation process converts any input tensor with the size $B \times C \times H \times W$ to $B \times C \times 1 \times H \times W$ by introducing an extra temporal axis. Secondly, as illustrated in Fig. 3, we extend the original transformer block to a Spatio-Temporal Transformer (ST-Transformer) by including a temporal attention layer after each spatial layer. Furthermore, we incorporate the concept of Rotary Positional Encoding (RoPE) from the recent LLM (Touvron et al., 2023) to integrate the temporal attention layer. Unlike previous methods that introduce an additional Temporal Transformer to model time, our modification directly applies to the transformer block itself, resulting in a simpler yet effective approach. Through various experiments with different designs of the temporal module, such as spatio-temporal attention and temporal causal attention, we observed that increasing the complexity of the temporal module only marginally improved the results while significantly increasing model size and training time. Therefore, we opt to retain the simplest design of the network, generating videos with 16 frames at a resolution of $320 \times 512$.

The primary objective of the base model is to generate high-quality key frames while also preserving diversity and capturing the compositional nature of videos. We aim to enable our model to synthesize videos aligned with creative prompts, such as *"Cinematic shot of Van Gogh's selfie"*. However, we observed that fine-tuning solely on video datasets, even with the initialization from a pre-trained LDM, fails to achieve this goal due to the phenomenon of catastrophic forgetting, where previous knowledge is rapidly forgotten after training for a few epochs. Hence, we apply a joint fine-tuning approach using both image and video data to address this issue. In practise, we concatenate $M$ images along the temporal axis to form a $T$-frame video and train the entire base model to optimize the objectives of both the Text-to-Image (T2I) and Text-to-Video (T2V) tasks (as shown in Fig. 3 (c)). Consequently, our training objective consists of two components: a video loss $\mathcal{L}_V$ and an image loss

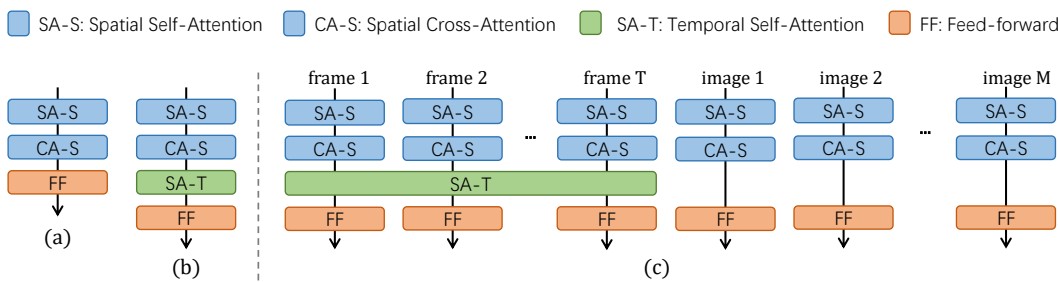

Figure 3: **Spatio-temporal module.** We show the Transformer block in Stable Diffusion in (a), our proposed ST-Transformer block in (b), and our joint image-video training scheme in (c).

$\mathcal{L}_I$. The overall objective can be formulated as:

$$\mathcal{L} = \mathbb{E}\left[\|\epsilon - \epsilon_\theta(\mathcal{E}(\mathbf{v}_t), t, c_V)\|_2^2\right] + \alpha * \mathbb{E}\left[\|\epsilon - \epsilon_\theta(\mathcal{E}(\mathbf{x}_t), t, c_I)\|_2^2\right], \tag{3}$$

where $c_V$ and $c_I$ represent the text descriptions for videos and images, respectively, and $\alpha$ is the coefficient used to balance the two losses. By incorporating images into the fine-tuning process, we observe a significant improvement in video quality. Furthermore, as demonstrated in Fig. 4, our approach successfully transfers various concepts from images to videos, including different styles, scenes, and characters. An additional advantage of our method is that, since we do not modify the architecture of LDM and jointly train on both image and video data, the resulting base model is capable of handling both T2I and T2V tasks, thereby showcasing the generalizability of our proposed design.

## 3.2 TEMPORAL INTERPOLATION MODEL

Building upon our base video latent diffusion model, we introduce a temporal interpolation network to enhance the smoothness of our generated videos and synthesize richer temporal details. We accomplish this by training a diffusion UNet, designed specifically to quadruple the frame rate of the base video. This network takes a 16-frame base video as input and produces an upsampled output consisting of 61 frames. During the training phase, we duplicate the base video frames to match the target frame rate and concatenate them with the noisy high-frame-rate frames. This combined data is used to train the diffusion UNet, enabling it to learn the process of denoising and generate the interpolated frames. At inference time, the base video frames are concatenated with randomly initialized Gaussian noise. The diffusion UNet gradually removes this noise through the denoising process, resulting in the generation of the 61 interpolated frames. Notably, our approach differs from conventional video frame interpolation methods, as each frame generated through interpolation replaces the corresponding input frame. In other words, every frame in the output is newly synthesized, providing a distinct approach compared to techniques where the input frames remain unchanged during interpolation. Furthermore, our diffusion UNet is conditioned on the text prompt, which serves as additional guidance for the temporal interpolation process, enhancing the overall quality and coherence of the generated videos.

## 3.3 VIDEO SUPER RESOLUTION MODEL

To further enhance visual quality and elevate spatial resolution, we incorporate a video super-resolution (VSR) model into our video generation pipeline. This involves training a LDM upsampler, specifically designed to increase the video resolution to $1280 \times 2048$. Similar to the base model described in Sec. 3.1, we leverage a pre-trained diffusion-based image $\times 4$ upscaler as a prior[1]. To adapt the network architecture to process video inputs in 3D, we incorporate an additional temporal dimension, enabling temporal processing within the diffusion UNet. Within this network, we introduce temporal layers, namely temporal attention and a 3D convolutional layer, alongside the existing spatial layers. These temporal layers contribute to enhancing temporal coherence in the generated videos. By concatenating the low-resolution input frames within the latent space, the diffusion UNet takes into account additional text descriptions and noise levels as conditions, which allows for more flexible control over the texture and quality of the enhanced output.

---

[1] https://huggingface.co/stabilityai/stable-diffusion-x4-upscaler

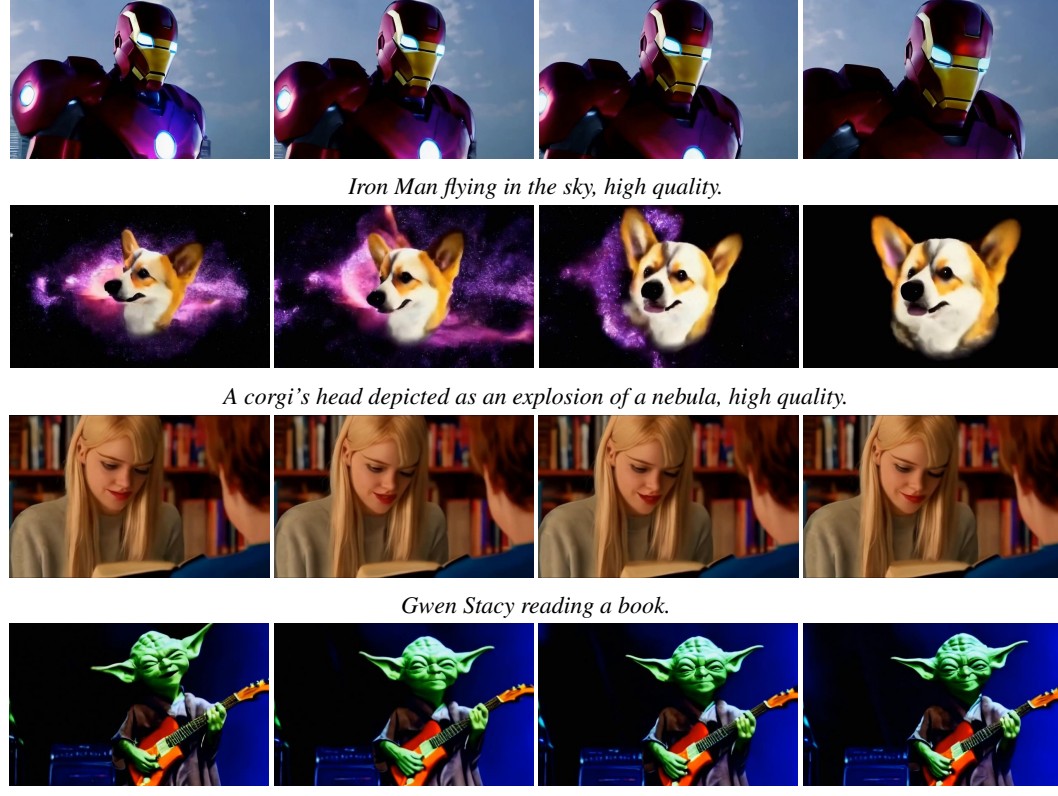

*Iron Man flying in the sky, high quality.*

*A corgi's head depicted as an explosion of a nebula, high quality.*

*Gwen Stacy reading a book.*

*Yoda playing guitar on the stage.*

Figure 4: **Diverse video generation results.** We show more videos from our method to demonstrate the diversity of our generated samples.

While the spatial layers in the pre-trained upscaler remain fixed, our focus lies in fine-tuning the inserted temporal layers in the V-LDM. Inspired by CNN-based super-resolution networks (Chan et al., 2022a;b; Zhou et al., 2022b; 2020; Jiang et al., 2021a; 2022), our model undergoes patch-wise training on $320 \times 320$ patches. By utilizing the low-resolution video as a strong condition, our upscaler UNet effectively preserves its intrinsic convolutional characteristics. This allows for efficient training on patches while maintaining the capability to process inputs of arbitrary sizes. Through the integration of the VSR model, our LaVie framework generates high-quality videos at a 2K resolution ($1280 \times 2048$), ensuring both visual excellence and temporal consistency in the final output.

## 4  EXPERIMENTS

In this section, we first introduce datasets in our experiments. Subsequently, we evaluate our method both qualitatively and quantitatively, comparing it to state-of-the-art approaches on the zero-shot text-to-video task. Finally, we showcase two applications of our method: long video generation and personalized video synthesis. In addition, we provide detailed implementation details in App. C, an in-depth analysis regarding the efficacy of joint image-video fine-tuning in App. D and show limitations of current approach in App. E.

### 4.1  DATASETS

To train our models, we leverage two publicly available datasets, namely Webvid10M (Bain et al., 2021) and Laion5B (Schuhmann et al., 2022). However, we encountered limitations when uitilizing WebVid10M for high-definition video generation, specifically regarding video resolution, diversity, and aesthetics. Therefore, we curate a new dataset called Vimeo25M, specifically designed to enhance the quality of text-to-video generation. By applying rigorous filtering criteria based on

resolution and aesthetic scores, we obtained a total of 20 million videos and 400 million images for training purposes.

The **Vimeo25M dataset** is a collection of 25 million text-video pairs in high-definition, widescreen, and watermark-free formats. These pairs are automatically generated using Videochat (Li et al., 2023). The original videos are sourced from Vimeo[2] and are classified into ten categories: *Ads and Commercials*, *Animation*, *Branded Content*, *Comedy*, *Documentary*, *Experimental*, *Music*, *Narrative*, *Sports*, and *Travel*. Example videos are shown in Fig. 7. To obtain the dataset, we utilized PySceneDetect[3] for scene detection and segmentation of the primary videos. To ensure the quality of captions, we filtered out captions with fewer than three words and excluded video segments with fewer than 16 frames. Consequently, we obtained a total of 25 million individual video segments, each representing a single scene. More detailed dataset statistics are introduced in App. B.

## 4.2 QUALITATIVE ANALYSIS

We present qualitative results of our approach, LaVie, through diverse text descriptions illustrated in Fig. 4. LaVie demonstrates its capability to synthesize videos with a wide range of content, including animals, movie characters, and various objects. Notably, our model exhibits a strong ability to combine spatial and temporal concepts, as exemplified by the synthesis of actions like *"Yoda playing guitar"*, which do not exist in the training set. These results indicate that our model learns to compose different concepts by capturing the underlying distribution rather than simply memorizing the training data. We show more results in Fig. 12.

Furthermore, we compare our generated results with three state-of-the-art and showcases the visual quality comparison in Fig. 5. LaVie outperforms Make-A-Video in terms of visual fidelity. Regarding the synthesis in the *"Van Gogh style"*, we observe that LaVie captures the style more effectively than the other two approaches. We attribute this to two factors: 1) initialization from a pretrained LDM facilitates the learning of spatio-temporal joint distribution, and 2) joint image-video fine-tuning mitigates catastrophic forgetting observed in Video LDM and enables knowledge transfer from images to videos more effectively. However, due to the unavailability of the testing code for the other two approaches, conducting a systematic and fair comparison is challenging.

## 4.3 QUANTITATIVE EVALUATION

We perform a zero-shot quantitative evaluation on two benchmark datasets, UCF101 (Soomro et al., 2012) and MSR-VTT (Chen et al., 2021), to compare our approach with existing methods. However, due to the time-consuming nature of sampling a large number of high-definition videos (e.g., ∼10000) using diffusion models, we limit our evaluation to using videos from the base models to reduce computational duration. Additionally, we observed that current evaluation metrics FVD may not fully capture the real quality of the generated videos. Therefore, to provide a comprehensive assessment, we conduct a large-scale human evaluation to compare the performance of our approach with state-of-the-art.

**Zero-shot Evaluation on UCF101.** We evaluate the quality of the synthesized results on UCF-101 dataset using the FVD, following the approach of TATS by employing the pretrained I3D (Carreira & Zisserman, 2017) model as the backbone. Similar to the methodology proposed in Video LDM, we utilize class names as text prompts and generate 100 samples per class, resulting in a total of 10,100 videos. During video sampling and evaluation, we generate 16 frames per video with a resolution of $320 \times 512$. Each frame is then center-cropped to a square size of $270 \times 270$ and resized to $224 \times 224$ to fit the I3D model input requirements.

The results, presented in Tab.1, demonstrate that our model outperforms all baseline methods, except for Make-A-Video. However, it is important to note that we utilize a smaller training dataset (WebVid10M+Vimeo25M) compared to Make-A-Video, which employs WebVid10M and HD-VILA-100M for training. Furthermore, in contrast to Make-A-Video, which manually designs a template sentence for each class, we directly use the class name as the text prompt, following the approach of Video LDM. When considering methods with the same experimental setting, our approach outperforms the state-of-the-art result of Video LDM by 24.31, highlighting the superiority of our method and underscoring the importance of the proposed dataset for zero-shot video generation.

---

[2] https://vimeo.com
[3] https://github.com/Breakthrough/PySceneDetect

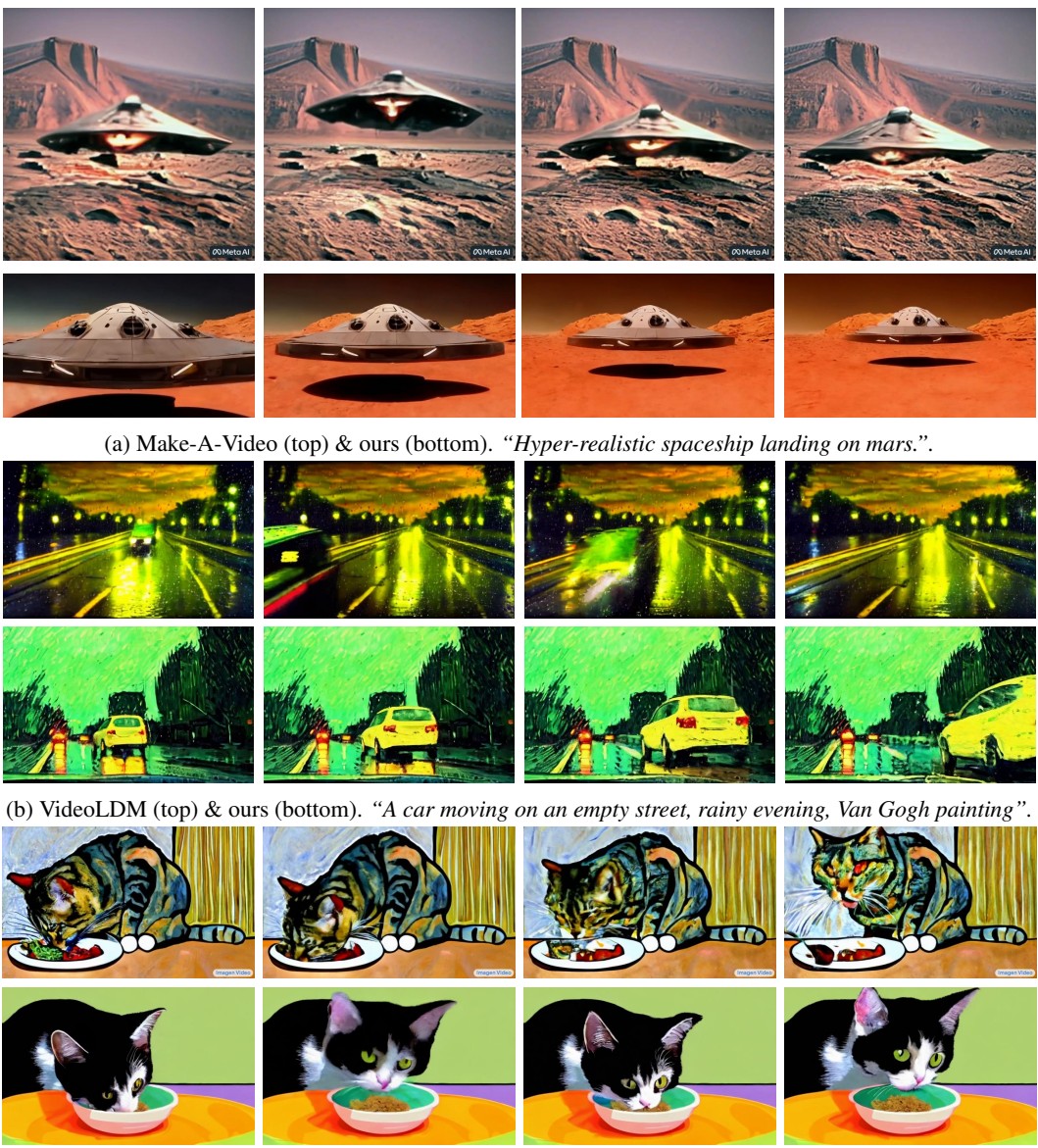

(a) Make-A-Video (top) & ours (bottom). *"Hyper-realistic spaceship landing on mars."*.

(b) VideoLDM (top) & ours (bottom). *"A car moving on an empty street, rainy evening, Van Gogh painting"*.

(c) Imagen Video (top) & ours (bottom). *"A cat eating food out of a bowl in style of Van Gogh"*.

Figure 5: **Comparison with state-of-the-art.** We compared to (a) Make-A-Video, (b) Video LDM and (c) Imagen Video. In each sub-figure, *bottom row* shows our result. We compare with Make-A-Video at spatial-resolution $512 \times 512$ and with the other two methods at $320 \times 512$.

Table 1: Comparison with SoTA *w.r.t.* FVD for zero-shot T2V generation on UCF101.

| Methods | Pretrain on image | Image generator | Resolution | FVD ($\downarrow$) |
|---|---|---|---|---|
| CogVideo (Chinese) (Hong et al., 2023) | No | CogView | $480 \times 480$ | 751.34 |
| CogVideo (English) (Hong et al., 2023) | No | CogView | $480 \times 480$ | 701.59 |
| Make-A-Video (Singer et al., 2023) | No | DALL·E2 | $256 \times 256$ | 367.23 |
| VideoFusion (Luo et al., 2023) | Yes | DALL·E2 | $256 \times 256$ | 639.90 |
| Magic Video (Zhou et al., 2022a) | Yes | Stable Diffusion | $256 \times 256$ | 699.00 |
| LVDM (He et al., 2022) | Yes | Stable Diffusion | $256 \times 256$ | 641.80 |
| Video LDM (Blattmann et al., 2023) | Yes | Stable Diffusion | $320 \times 512$ | 550.61 |
| Ours (w/o Vimeo25M) | Yes | Stable Diffusion | $320 \times 512$ | 540.30 |
| Ours | Yes | Stable Diffusion | $320 \times 512$ | 526.30 |

Table 2: Comparison with SoTA *w.r.t.* CLIPSIM for zero-shot T2V generation on MSR-VTT.

| Methods | Zero-Shot | CLIPSIM (↑) |
|---|---|---|
| GODIVA (Wu et al., 2021) | No | 0.2402 |
| NÜWA (Wu et al., 2022) | No | 0.2439 |
| CogVideo (Chinese) (Hong et al., 2023) | Yes | 0.2614 |
| CogVideo (English) (Hong et al., 2023) | Yes | 0.2631 |
| Make-A-Video (Singer et al., 2023) | Yes | 0.3049 |
| Video LDM (Blattmann et al., 2023) | Yes | 0.2929 |
| ModelScope (Wang et al., 2023b) | Yes | 0.2930 |
| Ours | Yes | 0.2949 |

**Zero-shot Evaluation on MSR-VTT.** For the MSR-VTT dataset, we conduct our evaluation by randomly selecting one caption per video from the official test set, resulting in a total of 2,990 videos. We assess the text-video semantic similarity using the clip similarity (CLIPSIM) metric. To compute CLIPSIM, we calculate the clip text-image similarity for each frame, considering the given text prompts, and then calculate the average score. In this evaluation, we employ the ViT-B-32 clip model as the backbone, following the methodology outlined in previous work (Blattmann et al., 2023) to ensure a fair comparison. Our experimental setup and details are consistent with the previous work. The results demonstrate that LaVie achieves superior or competitive performance compared to state-of-the-art methods, highlighting the effectiveness of our proposed training scheme and the utilization of the Vimeo25M dataset. These findings underscore the efficacy of our approach in capturing text-video semantic similarity.

**Human Evaluation.** Deviating from previous methods that primarily focus on evaluating general video quality, we contend that a more nuanced assessment is necessary to comprehensively evaluate the generated videos from various perspectives. In light of this, we compare our method with two existing approaches, VideoCrafter and ModelScope, leveraging the accessibility of their testing platforms. To conduct a thorough evaluation, we enlist the assistance of 30 human raters and employ two types of assessments. Firstly, we ask the raters to compare pairs of videos in three different scenarios: ours *v.s.* ModelScope, ours *v.s.* VideoCrafter, and ModelScope *v.s.* VideoCrafter. Raters are instructed to evaluate the overall video quality to vote which video in the pair has better quality. Secondly, we request raters to evaluate each video individually using five pre-defined metrics: motion smoothness, motion reasonableness, subject consistency, background consistency, and face, body, and hand quality. Raters are required to assign one of three labels, "good", "normal", or "bad" for each metric. All human studies are conducted without time limitations.

As presented in Tab. 3 and Tab. 4, our proposed method surpasses the other two approaches, achieving the highest preference among human raters. However, it is worth noting that all three approaches struggle to achieve a satisfactory score in terms of "motion smoothness" indicating the ongoing challenge of generating coherent and realistic motion. Furthermore, producing high-quality face, body, and hand visuals remains challenging.

Table 3: **Human Preference** on video quality.

| Metrics | Ours > ModelScope | Ours > VideoCrafter | ModelScope > VideoCrafter |
|---|---|---|---|
| Video quality | 75.00% | 75.58% | 59.10% |

Table 4: **Human Evaluation** on five pre-defined metrics. Each number signifies the proportion of examiners who voted for a particular category (good, normal, or bad) out of all votes.

| Metrics | VideoCraft | | | ModelScope | | | Ours | | |
|---|---|---|---|---|---|---|---|---|---|
| | Bad | Normal | Good | Bad | Normal | Good | Bad | Normal | Good |
| Motion Smoothness | 0.24 | 0.58 | 0.18 | 0.16 | 0.53 | 0.31 | 0.20 | 0.45 | **0.35** |
| Motion Reasonableness | 0.53 | 0.33 | 0.14 | 0.37 | 0.40 | 0.22 | 0.40 | 0.32 | **0.27** |
| Subject Consistency | 0.25 | 0.40 | 0.35 | 0.18 | 0.34 | 0.48 | 0.15 | 0.26 | **0.58** |
| Background Consistency | 0.10 | 0.40 | 0.50 | 0.08 | 0.28 | 0.63 | 0.06 | 0.22 | **0.72** |
| Face/Body/Hand quality | 0.69 | 0.24 | 0.06 | 0.51 | 0.31 | 0.18 | 0.46 | 0.30 | **0.24** |

## 4.4 MORE APPLICATIONS

In this section, we present two applications to showcase the capabilities of our pretrained models in downstream tasks: 1) long video generation, and 2) personalized video generation using LaVie.

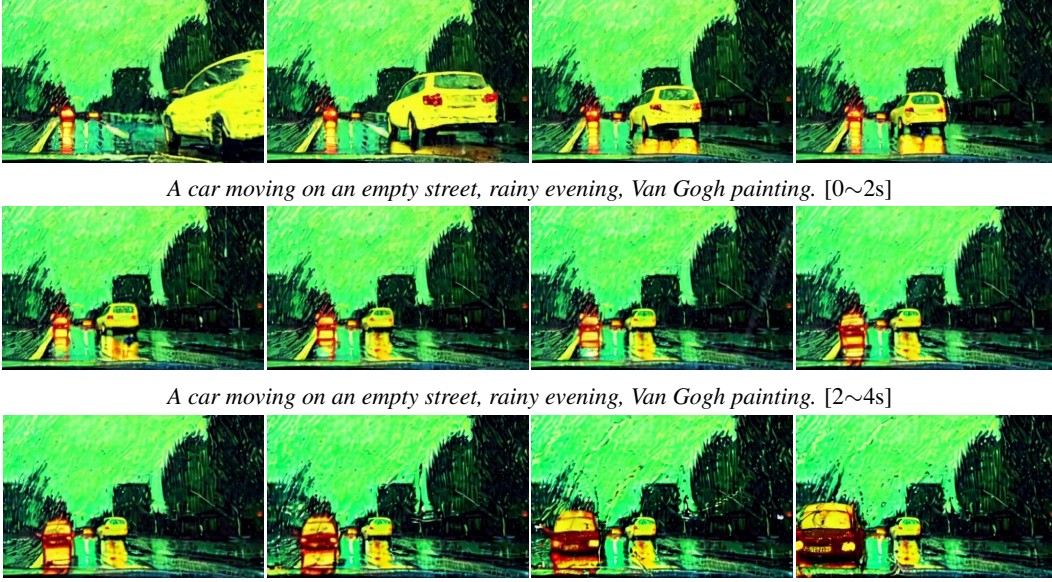

*A car moving on an empty street, rainy evening, Van Gogh painting.* [0∼2s]

*A car moving on an empty street, rainy evening, Van Gogh painting.* [2∼4s]

*A car moving on an empty street, rainy evening, Van Gogh painting.* [4∼6s]

Figure 6: **Long video generation.** By employing autoregressive generation three times consecutively, we successfully extend the video length of our base model from 2s to 6s.

**Long video generation.** To extend the video generation beyond a single sequence, we propose a simple recursive method. Similar to the temporal interpolation network, we incorporate the first frame of a video into the input layer of a UNet. By fine-tuning the base model accordingly, we enable the utilization of the last frame of the generated video as a conditioning input during inference. This recursive approach allows us to generate an extended video sequence. Fig. 6 showcases the results of generating tens of video frames (excluding frame interpolation) using this recursive manner, applied five times. The results demonstrate that the quality of the generated video remains high, with minimal degradation in video quality. This reaffirms the effectiveness of our base model in generating visually appealing frames.

**Personalized video generation.** Although our approach is primarily designed for general text-to-video generation, we demonstrate its versatility by adapting it to personalized video data through the integration of a personalized image generation approach, such as LoRA (Hu et al., 2022). In this adaptation, we fine-tune the spatial layers of our model using LoRA on self-collected images, while keeping the temporal modules frozen. As depicted in Fig. 14, the personalized video model for *"Misaka Mikoto"* is created after the fine-tuning process. The model is capable of synthesizing personalized videos based on various prompts. For instance, by providing the prompt *"Misaka Mikoto walking in the city"*, the model successfully generates scenes where *"Misaka Mikoto"* is depicted in novel places.

## 5 CONCLUSION

In this paper, we present **LaVie**, a text-to-video foundation model that produces high-quality and temporally coherent results. Our approach leverages a cascade of video diffusion models, extending a pre-trained LDM with simple designed temporal modules enhanced by Rotary Position Encoding (RoPE). To facilitate the generation of high-quality and diverse content, we introduce Vimeo25M, a novel and extensive video-text dataset that offers higher resolutions and improved aesthetics scores. By jointly fine-tuning on both image and video datasets, LaVie demonstrates a remarkable capacity to compose various concepts, including styles, characters, and scenes. We conduct comprehensive quantitative and qualitative evaluations for zero-shot text-to-video generation, which convincingly validate the superiority of our method over state-of-the-art approaches. Furthermore, we showcase the versatility of our pre-trained base model in long video generation and personalized video generation, which serve as additional evidence of the effectiveness and flexibility of LaVie.

## ETHIC STATEMENT

We acknowledge the ethical concerns that are shared with other T2I and T2V diffusion models. We aim to synthesize high-quality videos by giving text descriptions. Our approach can be used for movie production, making video games, artistic creation, generating synthetic data for other computer vision tasks, etc. We note that our framework has the potential to introduce unintended bias as a result of the training data.

## REPRODUCIBILITY STATEMENT

We intend to open-source our code, collected Vimeo25M dataset, as well as trained models.

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

# A  RELATED WORK

**Unconditional video generation** endeavors to generate videos by comprehensively learning the underlying distribution of the training dataset. Previous works have leveraged various types of deep generative models, including GANs (Goodfellow et al., 2016; Radford et al., 2015; Brock et al., 2019; Karras et al., 2019; 2020; Vondrick et al., 2016; Saito et al., 2017; Tulyakov et al., 2018; WANG et al., 2020; Wang et al., 2020; Wang, 2021; Wang et al., 2021; Clark et al., 2019; Brooks et al., 2022; Yu et al., 2022; Skorokhodov et al., 2022; Tian et al., 2021; Zhang et al., 2022; Dandi et al., 2020), VAEs (Kingma & Welling, 2014; Denton & Birodkar, 2017; Li & Mandt, 2018; Bhagat et al., 2020; Xie et al., 2020), and VQ-based models (Van Den Oord et al., 2017; Esser et al., 2021; Yan et al., 2021; Ge et al., 2022; Jiang et al., 2023). Recently, a notable advancement in video generation has been observed with the emergence of Diffusion Models (DMs) (Ho et al., 2020; Song et al., 2021a; Nichol & Dhariwal, 2021), which have demonstrated remarkable progress in image synthesis (Ramesh et al., 2021; 2022; Rombach et al., 2022). Building upon this success, several recent works (Ho et al., 2022b; He et al., 2022) have explored the application of DMs for video generation. These works showcase the promising capability of DMs to model complex video distributions by integrating spatio-temporal operations into image-based models, surpassing previous approaches in terms of video quality. However, learning the entire distribution of video datasets in an unconditional manner remains highly challenging. The entanglement of spatial and temporal content poses difficulties, making it still arduous to obtain satisfactory results.

**Text-to-video generation**, as a form of conditional video generation, focuses on the synthesis of high-quality videos using text descriptions as conditioning inputs. Existing approaches primarily extend text-to-image models by incorporating temporal modules, such as temporal convolutions and temporal attention, to establish temporal correlations between video frames. Notably, Make-A-Video (Singer et al., 2023) and Imagen Video (Ho et al., 2022a) are developed based on DALL·E2 (Ramesh et al., 2022) and Imagen (Saharia et al., 2022), respectively. PYoCo (Ge et al., 2023) proposed a noise prior approach and leverage a pre-trained eDiff-I (Balaji et al., 2022) as initialization. Conversely, other works (Blattmann et al., 2023; Zhou et al., 2022a; He et al., 2022) build upon Stable Diffusion (Rombach et al., 2022) owing to the accessibility of pre-trained models. In terms of training strategies, one approach involves training the entire model from scratch (Ho et al., 2022a; Singer et al., 2023) on both image and video data. Although this method can yield high-quality results by learning from both image and video distributions, it demands significant computational resources and entails lengthy optimization. Another approach is to construct the Text-to-Video (T2V) model based on pre-trained Stable Diffusion and subsequently fine-tune the model either entirely (Zhou et al., 2022a; He et al., 2022) or partially (Blattmann et al., 2023) on video data. These approaches aim to leverage the benefits of large-scale pre-trained T2I models to expedite convergence. However, we posit that relying exclusively on video data may not yield satisfactory results due to the substantial distribution gap between video and image datasets, potentially leading to challenges such as catastrophic forgetting. In contrast to prior works, our approach distinguishes itself by augmenting a pre-trained Stable Diffusion model with an efficient temporal module and jointly fine-tuning the entire model on both image and video datasets.

# B  VIMEO25M DATASET STATISTICS

The statistics of the Vimeo25M dataset, including the distribution of video categories, the duration of video segments, and the length of captions, are presented in Fig. 8. The dataset demonstrates a diverse range of categories, with a relatively balanced quantity among the majority of categories. Moreover, most videos in the dataset have captions consisting of approximately 10 words.

Furthermore, we conducted a comparison of the aesthetics score between the Vimeo25M dataset and the WebVid10M dataset. As illustrated in Fig. 9 (a), approximately 16.89% of the videos in Vimeo25M received a higher aesthetics score (greater than 6), surpassing the 7.22% in WebVid10M. In the score range between 4 and 6, Vimeo25M achieved a percentage of 79.12%, which is also superior to the 72.58% in WebVid10M. Finally, Fig. 9 (b) depicts a comparison of the spatial resolution between the Vimeo25M and WebVid10M datasets. It is evident that the majority of videos in the Vimeo25M dataset possess a higher resolution than those in WebVid10M, thereby ensuring that the generated results exhibit enhanced quality.

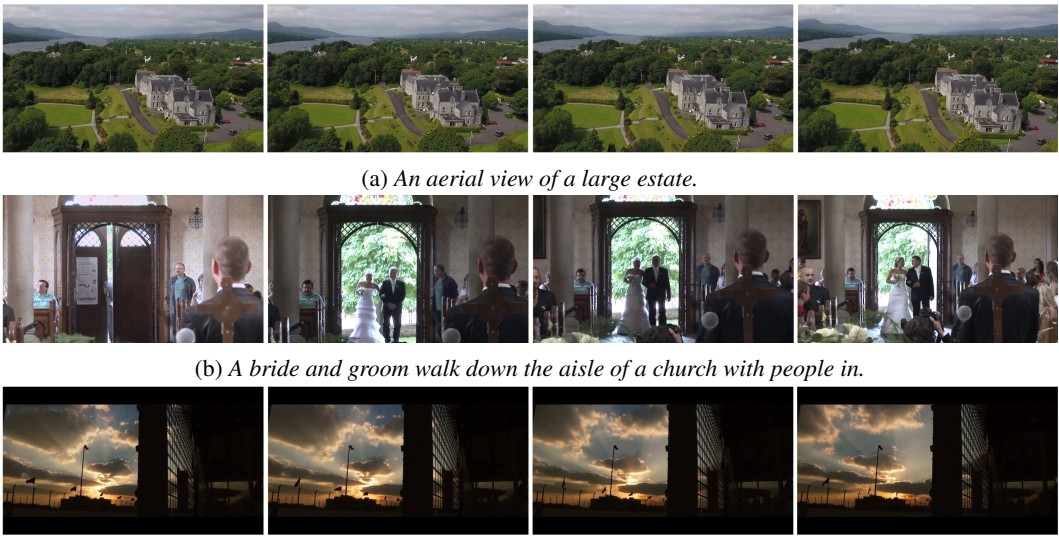

(a) *An aerial view of a large estate.*

(b) *A bride and groom walk down the aisle of a church with people in.*

(c) *A sunset with clouds in the sky.*

Figure 7: We show three video examples as well as text descriptions from Vimeo25M dataset.

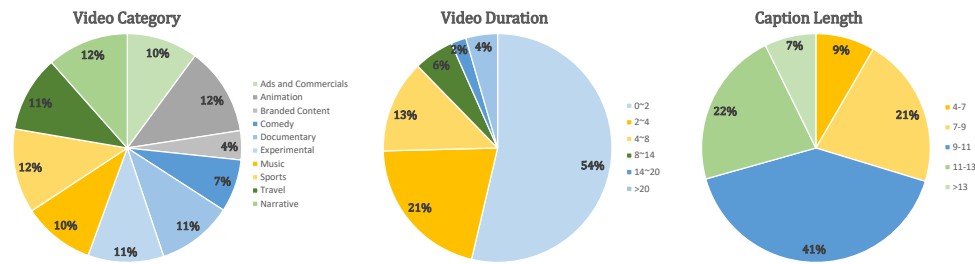

Figure 8: **Vimeo25M general information statistics.** We show statistics of video categories, clip durations, and caption word lengths in Vimeo25M.

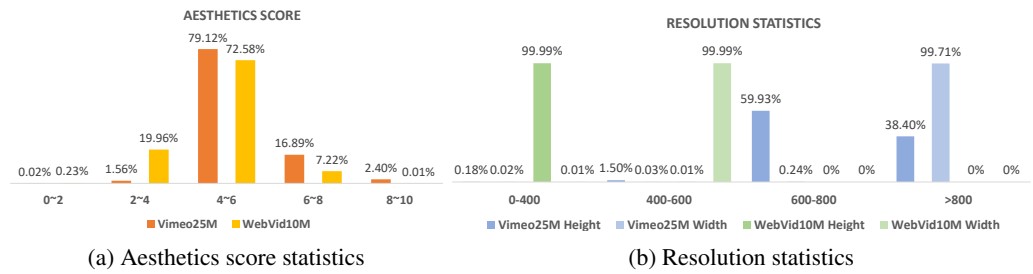

(a) Aesthetics score statistics        (b) Resolution statistics

Figure 9: **Aesthetics score, video scale statistics.** We compare Vimeo25M with WebVid10M in terms of (a) aesthetics score and (b) video spatial resolution.

## C   IMPLEMENTATION DETAILS

The Autoencoder and LDM of Base T2V model is initialized from a pretrained Stable Diffusion 1.4. Prior to training, we preprocess each video to a resolution of $320 \times 512$ and train using 16 frames per video clip. Additionally, we concatenate 4 images to each video for joint image-video fine-tuning. To facilitate the fine-tuning process, we employ curriculum learning (Bengio et al., 2009). In the initial stage, we utilize WebVid10M as the primary video data source, along with Laion5B, as the content within these videos is relatively simpler compared to the other dataset. Subsequently, we gradually introduce Vimeo25M to train the model on more complex scenes, subjects, and motion.

Temporal Interpolation model is initialized from our pretrained base T2V model. In order to accommodate our concatenated inputs of high and low frame-rate frames, we extend the architecture by incorporating an additional convolutional layer. During training, we utilize WebVid10M as the primary dataset. In the later stages of training, we gradually introduce Vimeo25M, which allows us to leverage its watermark-free videos, thus assisting in eliminating watermarks in the interpolated output. While patches of dimensions $256 \times 256$ are utilized during training, the trained model can successfully interpolate base videos at a resolution of $320 \times 512$ during inference.

The spatial layers of our VSR model is initialized from the pre-trained diffusion-based image ×4 upscaler, keeping these layers fixed throughout training. Only the newly inserted temporal layers, including temporal attention and 3D CNN layers, are trained. Similar to the base model, we employ the WebVid10M and Laion5B (with resolution $\geq$ 1024) datasets for joint image-video training. To facilitate this, we transform the image data into video clips by applying random translations to simulate handheld camera movements. For training purposes, all videos and images are cropped into patches of size $320 \times 320$. Once trained, the model can effectively process videos of arbitrary sizes, offering enhanced results.

## D    FURTHER ANALYSIS

In this section, we conduct a qualitative analysis of the training scheme employed in our experiments. We compare our joint image-video fine-tuning approach with two other experimental settings: 1) fine-tuning the entire UNet architecture based on WebVid10M, and 2) training temporal modules while keeping the rest of the network frozen. The results, depicted in Fig. 10, highlight the advantages of our proposed approach. When fine-tuning the entire model on video data, we observe a phenomenon known as catastrophic forgetting. The concept of *"teddy bear"* gradually diminishes and the quality of its representation deteriorates significantly. Since the training videos contain very few instances of *"teddy bear"*, the model gradually adapts to the new data distribution, resulting in a loss of prior knowledge. In the second setting, we encounter difficulties in aligning the spatial knowledge from the image dataset with the newly learned temporal information from the video dataset. The significant distribution gap between the image and video datasets poses a challenge in effectively integrating the spatial and temporal aspects. The attempts made by the high-level temporal modules to modify the spatial distribution adversely affect the quality of the generated videos. In contrast, our proposed joint image-video fine-tuning approach effectively learns the joint distribution of image and video data. This enables the model to recall knowledge from the image dataset and apply the learned motion from the video dataset, resulting in higher-quality synthesized videos. The ability to leverage both datasets enhances the overall performance and quality of the generated results.

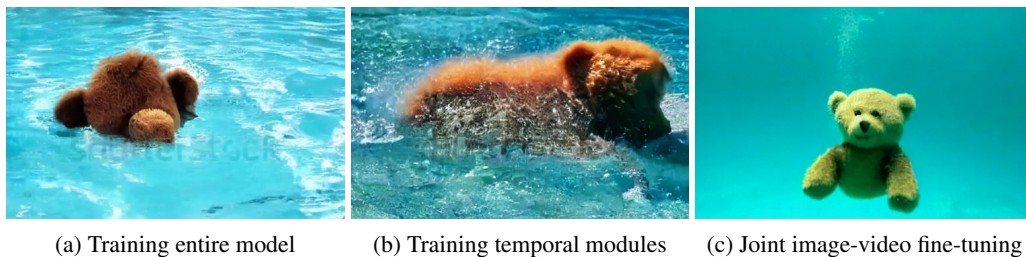

(a) Training entire model          (b) Training temporal modules          (c) Joint image-video fine-tuning

Figure 10: **Training scheme comparison.** We show image results based on (a) training the entire model, (b) training temporal modules, and (c) joint image-video fine-tuning, respectively.

## E    LIMITATIONS

While LaVie has demonstrated impressive results in general text-to-video generation, we acknowledge the presence of certain limitations. In this section, we highlight two specific challenges which are shown in Fig. 11:

*Multi-subject generation:* Our models encounter difficulties when generating scenes involving more than two subjects, such as *"Albert Einstein discussing an academic paper with Spiderman"*. There are instances where the model tends to mix the appearances of Albert Einstein and Spiderman,

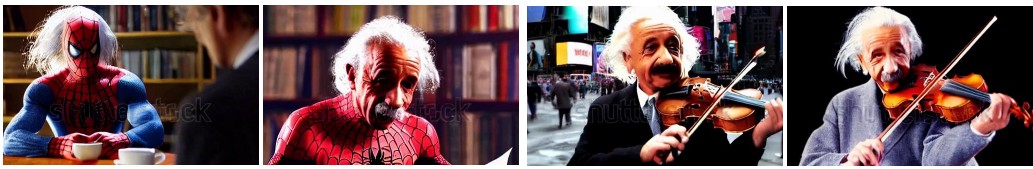

(a) *Albert Einstein discussing an academic paper with Spiderman.*

(b) *Albert Einstein playing the violin.*

Figure 11: **Limitations.** We show limitations on (a) mutiple-object generation and (b) failure of hands generation

instead of generating distinct individuals. We have observed that this issue is also prevalent in the T2I model (Rombach et al., 2022). One potential solution for improvement involves replacing the current language model, CLIP (Radford et al., 2021), with a more robust language understanding model like T5 (Raffel et al., 2020). This substitution could enhance the model's ability to accurately comprehend and represent complex language descriptions, thereby mitigating the mixing of subjects in multi-subject scenarios.

*Hands generation:* Generating human bodies with high-quality hands remains a challenging task. The model often struggles to accurately depict the correct number of fingers, leading to less realistic hand representations. A potential solution to address this issue involves training the model on a larger and more diverse dataset containing videos with human subjects. By exposing the model to a wider range of hand appearances and variations, it could learn to generate more realistic and anatomically correct hands.

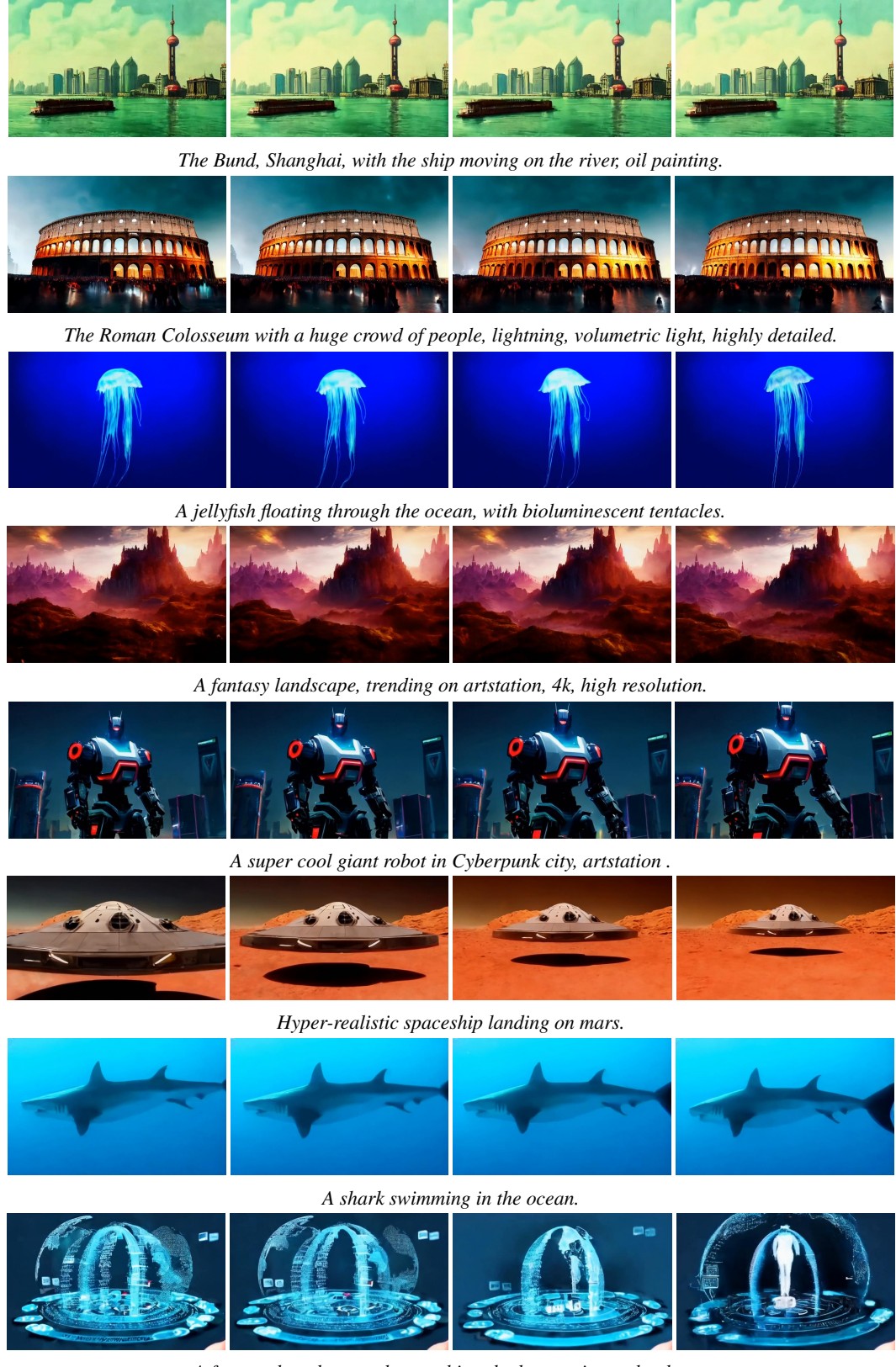

*The Bund, Shanghai, with the ship moving on the river, oil painting.*

*The Roman Colosseum with a huge crowd of people, lightning, volumetric light, highly detailed.*

*A jellyfish floating through the ocean, with bioluminescent tentacles.*

*A fantasy landscape, trending on artstation, 4k, high resolution.*

*A super cool giant robot in Cyberpunk city, artstation .*

*Hyper-realistic spaceship landing on mars.*

*A shark swimming in the ocean.*

*A future where humans have achieved teleportation technology .*

Figure 12: **Diverse video generation results.**

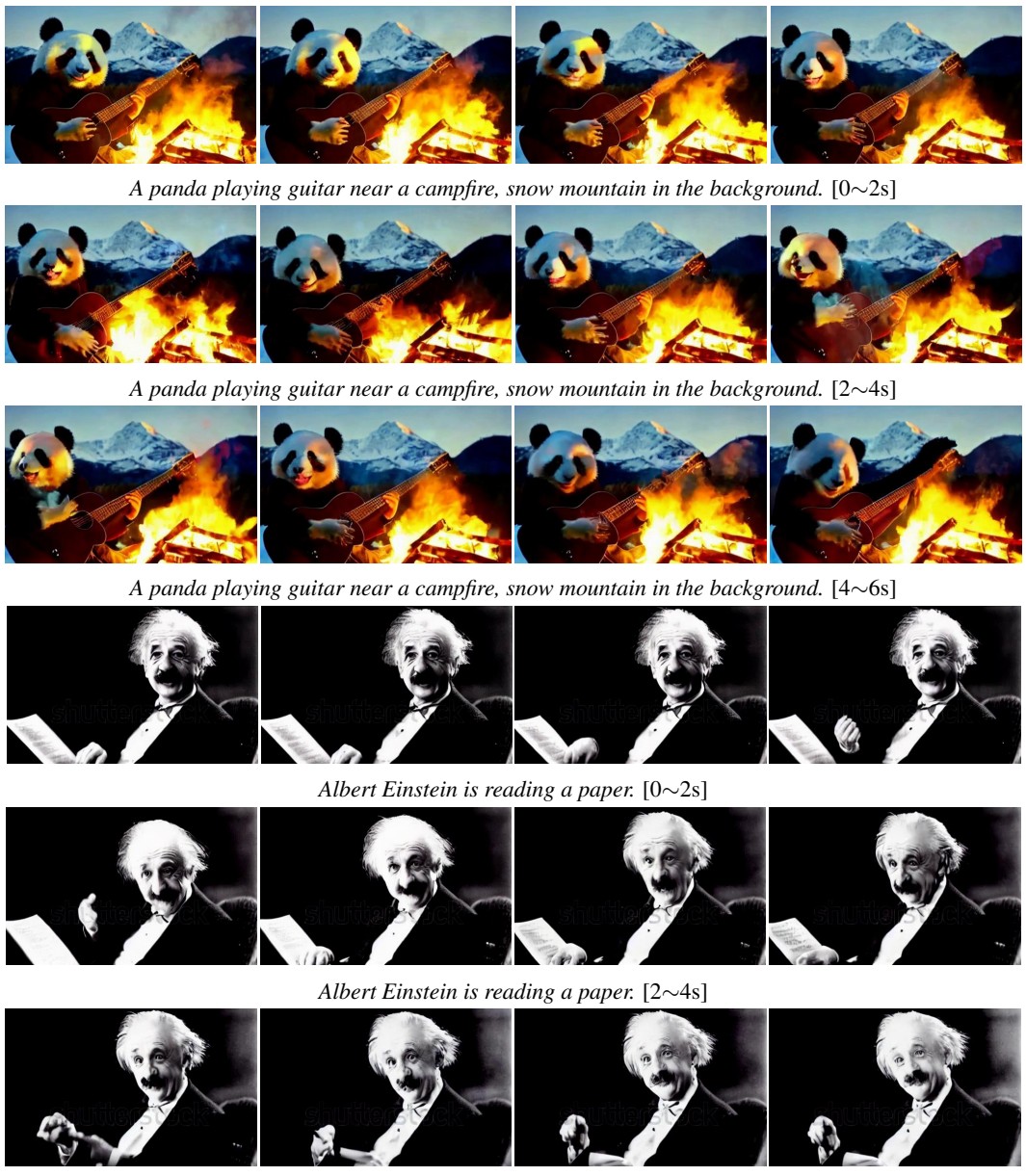

*A panda playing guitar near a campfire, snow mountain in the background.* [0∼2s]

*A panda playing guitar near a campfire, snow mountain in the background.* [2∼4s]

*A panda playing guitar near a campfire, snow mountain in the background.* [4∼6s]

*Albert Einstein is reading a paper.* [0∼2s]

*Albert Einstein is reading a paper.* [2∼4s]

*Albert Einstein is reading a paper.* [4∼6s]

Figure 13: **Long video generation.** By employing autoregressive generation three times consecutively, we successfully extend the video length of our base model from 2s to 6s.

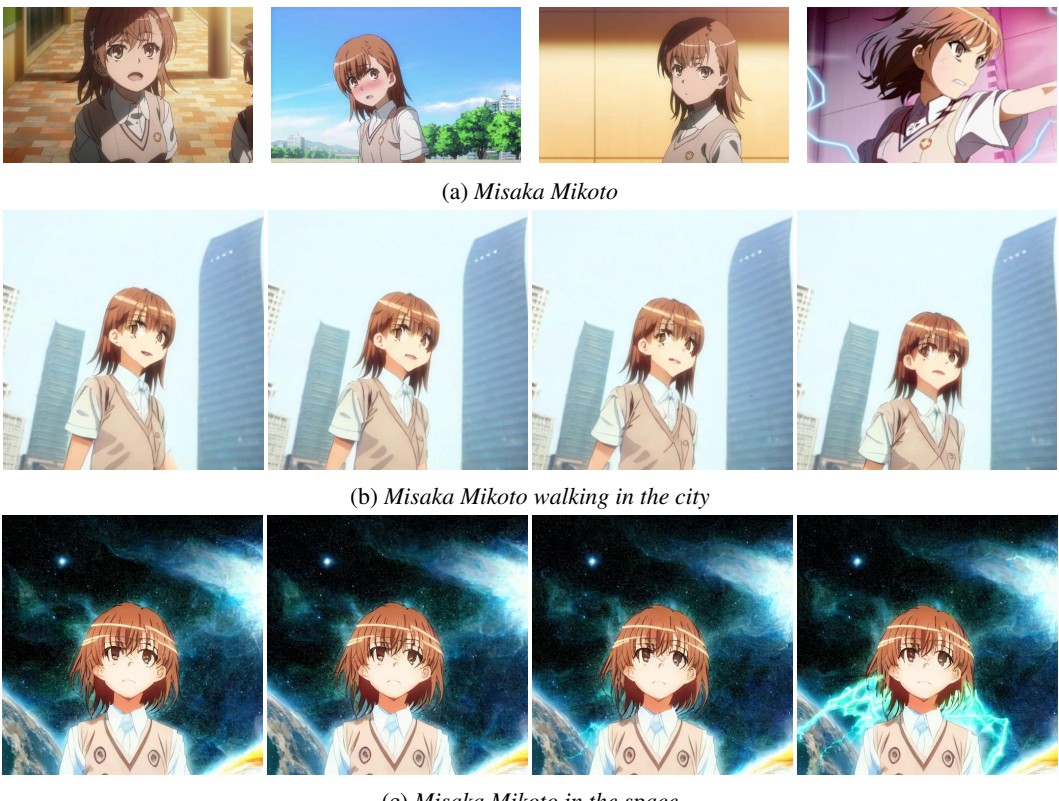

(a) *Misaka Mikoto*

(b) *Misaka Mikoto walking in the city*

(c) *Misaka Mikoto in the space*

Figure 14: **Personalized video generation.** We show results by adopting a LoRA-based approach in our model for personalized video generation. Samples used to train our LoRA are shown in (a). We use "Misaka Mikoto" as text prompts. Results from our video LoRA are shown in (b) and (c). By inserting pre-trained temporal modules into LoRA, we are able to animate "Misaka Mikoto" and control the results by combining them with different prompts.

