# OpenReview forum: "LaVie: High-Quality Video Generation with Cascaded Latent Diffusion Models"
_ICLR.cc/2024/Conference — Submitted to ICLR 2024_

### Official Review · Reviewer_9GLX · 2023-10-29

**Soundness:** 2 fair
**Presentation:** 2 fair
**Contribution:** 2 fair
**Rating:** 5
**Confidence:** 4

**Summary:**

This work introduces LaVie, a text-to-video generative model that builds upon a pre-trained text-to-image model.  LaVie consists of cascaded video latent diffusion models, including a base T2V model, a temporal interpolation model, and a video super-resolution model. The key insights include the use of temporal self-attentions and rotary positional encoding to capture temporal correlations in video data. Joint image-video fine-tuning is crucial for high-quality results. The authors also contribute a large and diverse video dataset called Vimeo25M. Experimental results demonstrate that LaVie achieves state-of-the-art performance in both quantitative and qualitative measuress.

**Strengths:**

The article provides some insights:
1. The use of simple temporal self-attention mechanisms coupled with rotary positional encoding to capture temporal correlations in video data
2. Joint image-video fine-tuning in producing high-quality and creative outcomes.
3. The authors contribute the Vimeo25M dataset, which comprises 25 million text-video pairs, serving as a valuable resource for research and development in the field.

**Weaknesses:**

1. I think the article lacks technical innovation. Its core contribution involves extending the pre-trained LDM to a video generation model, introducing temporal attention, and emphasizing joint image-video training. However, similar techniques have been mentioned in previous works, such as VDM and Align your latent. The authors should clarify the distinguishing aspects of these innovation.

2. The article presents several assertions without strong empirical support or ablation study. For instance, the introduction of RoPE as a positional encoding lacks corresponding experimental evidence to demonstrate its effectiveness and advantages over other encoding methods. The statement, "we validate that the process of joint image-video fine-tuning plays a pivotal role," lacks experimental substantiation in the article.

3. The writing should be improved, and there exist some unclear explanations. For instance, the modules in Figure 2 are not adequately introduced in the text, causing confusion. For example,  I want to know whether 'E' denotes the encoder or denoiser? Additionally, the article mentions "By applying rigorous filtering criteria" to construct Vimeo25M, but the specific criteria are not outlined. Will this dataset be made publicly available?

**Questions:**

See Weaknesses.

---

> ### Author Response · Authors · 2023-11-22
> **Response from the Authors**
>
> **Q1. I think the article lacks technical innovation. Its core contribution involves extending the pre-trained LDM to a video generation model, introducing temporal attention, and emphasizing joint image-video training. However, similar techniques have been mentioned in previous works, such as VDM and Align your latent. The authors should clarify the distinguishing aspects of these innovation.**
>
> **A1.** we consider our contributions are system-level rather than model-level. We leverage a "base-interpolation-video super resolution" system design.
>
> For the entire system, we made the following contributions:
> - Model design: we propose a simple yet effective method to involve RoPE-equipped temporal self-attention in our base, interpolation and video super resolution models. Experiments show that such design is able to produce videos with temporal coherency.
> - Training: we studied different training schemes in base, interpolation and super-resolution models, and proved that when given a pre-trained text-to-image generation model, joint image-video fine-tuning is an effective way to overcome catatrophical forgeting, as well as to obtain high quality videos.
> - Evaluation: T2V is a novel video generation task, where we think novel evaluation is required. Different from other approaches by simply reporting human preference and FVD, we conduct a systematic human evaluation on various dimensions to compare with state-of-the-art.
> - Dataset: We propose a novel text-video dataset Vimeo25M to improve the performance of text-to-video generation tasks.
>
> **Q2. The article presents several assertions without strong empirical support or ablation study. For instance, the introduction of RoPE as a positional encoding lacks corresponding experimental evidence to demonstrate its effectiveness and advantages over other encoding methods. The statement, "we validate that the process of joint image-video fine-tuning plays a pivotal role," lacks experimental substantiation in the article.**
>
> **A2.** We have conducted small-scale experiments on UCF101 for comparison. We explore three different settings. 1)  Our proposed temporal module, 2) replacing RoPE in our module with absolute PE, 3) spatial-temporal self-attention without PE used in [1], We train three models using same iterations and report FVD in Table
>
>
> |Method|Ours|absolute PE|spatio-temporal self-attention|
> |----|----|----|----|
> |FVD|611|625|635|
>
> Small-scale experimental results demonstrate the effectiveness of our proposed temporal module.
> We show results with and without image-video joint fine-tuning in our updated supplementary material website. Results show that, without image-video joint tuning, catatrophical forgeting appear in the generated results. Such as the teddy bear and Iron Man could not be well generated, the entire system overfit to training video data.
>
> **Q3. The writing should be improved, and there exist some unclear explanations. For instance, the modules in Figure 2 are not adequately introduced in the text, causing confusion. For example, I want to know whether 'E' denotes the encoder or denoiser. Additionally, the article mentions "By applying rigorous filtering criteria" to construct Vimeo25M, but the specific criteria are not outlined. Will this dataset be made publicly available?**
>
> **A3.** Thanks to the reviewer for pointing it out. E represents Encoder in VAE. We have updated the figure, as well as the caption in the revised manuscript.
>
> The original videos are sourced from Vimeo. To obtain the dataset, we utilized PySceneDetect for scene detection and segmentation of the primary videos. We implemented camera motion detection and static video analysis primarily to filter out videos with camera shake and static footage. We retained only those segments where the camera was either stationary or moving gently, while segments with entirely static footage were eliminated. To ensure the quality of captions, we filtered out captions with fewer than three words and excluded video segments with fewer than 16 frames.
>
> Yes, Vimeo25M will be publicly available.
>
> [1] Wu et al, Tune-a-video: One-shot tuning of image diffusion models for text-to-video generation, CVPR 2023

---

### Official Review · Reviewer_aq4u · 2023-10-31

**Soundness:** 3 good
**Presentation:** 3 good
**Contribution:** 2 fair
**Rating:** 5
**Confidence:** 5

**Summary:**

This paper proposes a text-to-video method by using a pre-trained text-to-image model as the initialization. The authors present two key findings: 1) temporal self-attention with relative position encoding could maintain temporal consistency well. 2) joint image-video fine-tuning is crucial for good performance. This paper also contributes a video dataset named Vimeo25M, with 25 million text-video pairs.

**Strengths:**

1. The paper is well-written and easy to follow, and the visualization results seem appealing.
2. The paper proposes a new video-text datasets named Vimeo25M.

**Weaknesses:**

1. The technical novelty is limited. The authors present two findings which have both been proposed in previous works. 1) temporal attention over a pretrained text-to-image model could address the temporal consistency model well. This idea is first stated in “AnimateDiff: Animate Your Personalized Text-to-Image Diffusion Models without Specific Tuning” and then pointed out in "MagicEdit: High-Fidelity and Temporally Coherent Video Editing". 2) Joint image-video training is a well-known technique to improve the effectiveness of training text-to-video models. It was first proposed by Jonathan Ho et al. in "Video Diffusion Models".
2.  The idea of the cascaded diffusion model, which first trains a base model then temporally interpolates it, and finally performs super-resolution spatially, is a standard procedure to learn a text-to-video model. Similar ideas are also proposed in "Make-A-Video: Text-to-video Generation Without Text-Video Data", "Imagen Video: High Definition Video Generation with Diffusion Models" and "Align your Latents: High-Resolution Video Synthesis with Latent Diffusion Models".

**Questions:**

Will the authors release the newly proposed video dataset Vimeo25M? Since it seems that there is no promise from the authors that they will release the dataset. I would say that the major contribution of the paper is the dataset.

---

> ### Author Response · Authors · 2023-11-22
> **Response from the Authors**
>
> **Q1. The technical novelty is limited. The authors present two findings which have both been proposed in previous works. 1) temporal attention over a pretrained text-to-image model could address the temporal consistency model well. This idea is first stated in “AnimateDiff: Animate Your Personalized Text-to-Image Diffusion Models without Specific Tuning” and then pointed out in "MagicEdit: High-Fidelity and Temporally Coherent Video Editing". 2) Joint image-video training is a well-known technique to improve the effectiveness of training text-to-video models. It was first proposed by Jonathan Ho et al. in "Video Diffusion Models".**
>
> **A1.**
> 1) We think LaVie is a concurrent work compared to AnimateDiff and MagicEdit since both works have not been peer-reviewed and are only on arXiv.
>
> 2) Joint image-video training is a well-known technique for both video understanding [1] and video generation for a long time. In this work, we studied how to leverage this technique to fine-tune a pretrained Stable Diffusion initialized T2V system. We demonstrate that joint image-video fine-tuning is the key to obtaining good results for base, interpolation, as well as video super resolution models. We have also shown that such technique helps to prevent catastrophic forgeting. However, there are still open questions about using this technique, such as 1) is there any balance between image and video, and 2) how to also leverage images in temporal modules to make the best usage of the entire training data. Such questions are also related to design effective learning approaches in video diffusion models. Therefore, we consider that our contributions are not in this technique itself, but the analysis of this technique, as well as opening new doors for future usage and improvement.
>
> **Q2. The idea of the cascaded diffusion model, which first trains a base model then temporally interpolates it, and finally performs super-resolution spatially, is a standard procedure to learn a text-to-video model. Similar ideas are also proposed in "Make-A-Video: Text-to-video Generation Without Text-Video Data", "Imagen Video: High Definition Video Generation with Diffusion Models" and "Align your Latents: High-Resolution Video Synthesis with Latent Diffusion Models".**
>
> **A2.** We have already discussed the differences with listed works in the original manuscript in Related Works.
>
> Since the "base-interpolation-SR" framework has become a standard design for high-quality video generation, we consider our contributions are system-level rather than model-level.
>
> For the entire system, we made the following contributions:
> - Model design: we propose a simple yet effective method to involve RoPE-equipped temporal self-attention in our base, interpolation and video super resolution models. Experiments show that such design is able to produce videos with temporal coherency.
> - Training: we studied different training schemes in base, interpolation and super-resolution models, and proved that when given a pre-trained text-to-image generation model, joint image-video fine-tuning is an effective way to overcome catatrophical forgeting, as well as to obtain high quality videos.
> - Evaluation: T2V is a novel video generation task, where we think novel evaluation is required. Different from other approaches by simply reporting human preference and FVD, we conduct a systematic human evaluation on various dimensions to compare with state-of-the-art.
> - Dataset: We propose a novel text-video dataset Vimeo25M to improve the performance of text-to-video generation tasks.
>
> **Q3. Will the authors release the newly proposed video dataset Vimeo25M? Since it seems that there is no promise from the authors that they will release the dataset. I would say that the major contribution of the paper is the dataset.**
>
> **A3.** Yes, Vimeo25m will be released.
>
> [1] Bain et al., Frozen in Time: A Joint Video and Image Encoder for End-to-End Retrieval, ICCV 2021

---

### Official Review · Reviewer_b3WW · 2023-10-31

**Soundness:** 3 good
**Presentation:** 2 fair
**Contribution:** 3 good
**Rating:** 6
**Confidence:** 5

**Summary:**

The paper presents LaVie, a novel integrated video generation framework, which is fundamentally constructed on cascaded video latent diffusion models. The authors strategically integrate three principal components into the LaVie framework: a base Text-to-Video (T2V) model, a temporal interpolation model, and a video super-resolution model. Their approach is underscored by two primary insights: firstly, the integration of simplistic temporal self-attentions, when paired with rotary positional encoding, proves essential in capturing the temporal correlations intrinsic to video data efficiently. Secondly, the process of concurrently fine-tuning both image and video, is validated as being crucial in generating outcomes marked by high-quality and creativity.

**Strengths:**

1. Innovative Integrated Framework:
The paper presents LaVie, a cutting-edge integrated video generation framework with an enormous capacity of 3 billion parameters. LaVie, primarily a text-to-video foundation model, is meticulously constructed to synthesize visually appealing and temporally coherent videos. The model maintains the vigorous creative generation characteristics of a pre-trained Text-to-Image (T2I) model, ensuring synthesized videos are not only realistic but also infused with a strong creative essence.

2. Strategic Incorporation of Key Insights:
The authors introduce two pivotal insights that form the backbone of LaVie’s design. Firstly, a combination of simple temporal self-attention with Rotary Positional Encoding (RoPE) is utilized, proving sufficient in capturing the intrinsic temporal correlations present in video data. Secondly, the paper emphasizes the indispensable role of joint image-video fine-tuning in crafting high-quality and imaginative outcomes, ensuring the model retains and effectively utilizes learned prior knowledge without succumbing to catastrophic forgetting.

3. Introduction of a Comprehensive Dataset:
Recognizing the limitations of existing datasets, the authors contribute a novel text-video dataset named Vimeo25M. This dataset is a treasure trove of 25 million high-resolution videos, each accompanied by text descriptions, curated to overcome the prevalent issues of low-resolution and watermarked content found in previous datasets. The utilization of Vimeo25M significantly amplifies LaVie’s performance, empowering it to churn out results that excel in quality, diversity, and aesthetic allure, thus substantially advancing the Text-to-Video (T2V) synthesis task.

**Weaknesses:**

1. Some details are not clear. See **Questions** below.

2. Availability of Contributed Dataset:
A significant contribution highlighted in this paper is the introduction of the Vimeo25M dataset. However, it's crucial to clarify that, as of now, this dataset has not been made publicly available for utilization and further exploration.

3. The claim:

>  simple temporal self-attention coupled with RoPE (Su et al., 2021) adequately captures temporal
correlations inherent in video data. More complex architectural design only results in marginal visual improvements to the generated outcomes

isn't supported by experimental results.

**Questions:**

1. Positional Embedding:
Is the utilization of Rotary Positional Encoding (RoPE) distinctly advantageous compared to other available options, such as learnable embeddings? The clarity of the benefits associated with using RoPE, as opposed to its counterparts, would be appreciated.

2. Clarification on Image Concatenation:
The process involving the concatenation of M images along the temporal axis to formulate a T-frame video raises a query. Specifically, is there an equivalence between the variables `M` and  `T` in this context?

3. Query on SA-T Removal in Image Training:
Figure 3 (c) seems to imply that the SA-T is omitted during the image training phase. Could you provide a more comprehensive explanation or clarification regarding this aspect?

4. Interaction of Text Conditioning with Spatial Attention:
The paper suggests that text conditioning primarily interacts with spatial attention. Given this, how does text effectively influence or control motion within the model? More details regarding the interplay between text and motion would enhance understanding.

---

> ### Author Response · Authors · 2023-11-22
> **Response from the Authors**
>
> **Q1. Availability of Contributed Dataset: A significant contribution highlighted in this paper is the introduction of the Vimeo25M dataset. However, it's crucial to clarify that, as of now, this dataset has not been made publicly available for utilization and further exploration.**
>
> **A1.** Yes, the Vimeo25M dataset will be publicly available soon.
>
> **Q2. The claim: simple temporal self-attention coupled with RoPE (Su et al., 2021) adequately captures temporal correlations inherent in video data. More complex architectural design only results in marginal visual improvements to the generated outcomes isn't supported by experimental results.**
>
> **A2.**  For the temporal module, we have conducted small-scale experiments on UCF101 for comparison. We explore three different settings. 1)  Our proposed temporal module, 2) replacing RoPE in our module with absolute PE, 3) spatial-temporal self-attention used in [1], We train three models using same iterations and report FVD in Table
>
> |Method|Ours|absolute PE|spatio-temporal self-attention|
> |----|----|----|----|
> FVD|611|625|635|
>
> While 3) is much complex than our proposed temporal module,  we didn't find improvement in visual quality and training is slower. Small-scale quantitative evaluation demonstrates the effectiveness of our proposed temporal module.
>
> **Q3. Positional Embedding: Is the utilization of Rotary Positional Encoding (RoPE) distinctly advantageous compared to other available options, such as learnable embeddings? The clarity of the benefits associated with using RoPE, as opposed to its counterparts, would be appreciated.**
>
> **A3.** We have conducted experiments to compare RoPE with absolute PE on UCF101. Results show that we are able to obtain lower FVD using RoPE.
>
> In LLM, compared to absolute PE, RoPE is able to extropolate longer text length beyond training length. Our motivation is to adapt this property also for video generation, e.g., training on 16 frames and inference for 32 or even more frames. We show results in updated supplementary material website. We train on 16 frames, and generate 32, 48, 64 frames for each prompt. Results show that we are able to generate longer videos with good spatial quality. While our method is able to generate video length beyond training data, we found flickering becomes larger when the generated videos become longer. Hence, we believe that it is still an open question about how to design better PE for video generation.
>
> **Q4. Clarification on Image Concatenation: The process involving the concatenation of M images along the temporal axis to formulate a T-frame video raises a query. Specifically, is there an equivalence between the variables M and T in this context?**
>
> **A4.** In our experiments, we try to make the best use of our GPU memory towards adding as many images as we could for each video. Currently, we add T=4 images to each M=16 frames videos. Benchmarking M and T is hard at current stage as it requires conducting abalation experiments on large-scale datasets, which may require months computing. We will leave it in our future work to explore.
>
> **Q5. Query on SA-T Removal in Image Training: Figure 3 (c) seems to imply that the SA-T is omitted during the image training phase. Could you provide a more comprehensive explanation or clarification regarding this aspect?**
>
> **A5.** Yes, SA-T is omitted during the image training phase. The reason is, temporal self-attention is not able to be conducted on images since there is no temporal correlation between random concatenated images. Images are only used for optimizing spatial parts.
>
> **Q6. Interaction of Text Conditioning with Spatial Attention: The paper suggests that text conditioning primarily interacts with spatial attention. Given this, how does text effectively influence or control motion within the model? More details regarding the interplay between text and motion would enhance understanding.**
>
> **A6.** We have explored to add an extra temporal-level cross-attention for text input. However, we didn't observe huge differences in the generated results. The reason could be while text has been used in spatial-level cross-attention, it can still be considered as a type of condition for the entire video. When the entire spatio-temporal networks are optimized, such conditions will as well affect the temporal module, hence control motion.
>
> [1] Wu et al, Tune-a-video: One-shot tuning of image diffusion models for text-to-video generation, CVPR 2023

---

### Official Review · Reviewer_m4rL · 2023-11-03

**Soundness:** 3 good
**Presentation:** 3 good
**Contribution:** 2 fair
**Rating:** 6
**Confidence:** 4

**Summary:**

In this paper, the authors present a text-to-video model consisting of multiple components including text-2-video, temporal interpolation and super-resolution components. The text-2-video model is built on the pre-trained text-2-image model. Additionally, they introduce Vimeo25M dataset to enhance the quality of text-to-video generation. The method is straightforward and the paper showcases the versatility of the model in long video generation and personalized video generation.

**Strengths:**

- The paper is well-written and easy to follow

- The resulting model is capable of handling both T2I and T2V tasks.

- The paper provides a human evaluation of video generation quality.

- The paper introduces Vimeo25M dataset which is a collection of 25 million text-video pairs. The dataset aids in boosting the performance of model in terms of quality and diversity.

- Joint image-video training of model is interesting and seems a reasonable approach for training video generation models.

**Weaknesses:**

1- The technical novelty of the proposed method is limited as it is a combination of existing techniques, including text-to-image generation, frame interpolation, and super-resolution.

2- Dandi et al.'s work on “Jointly Trained Image and Video Generation using Residual Vectors" is relevant to this paper's exploration of joint image-video training. However, this paper is neither cited nor discussed.

3- There is no analysis of the contribution of individual components or techniques on the video generation performance. The paper mentions the temporal module, joint image-video training, and usage of Vimeo25M dataset for training. However, we don’t see a comprehensive analysis on the impact of each of them on the performance. It’s hard to understand which component has a higher impact on the final model. The only analysis we see is in Fig 10 which is very limited by showing only three images.

4- The role of the temporal self-attention module (SA-T) during image-only training phases is ambiguous. It's unclear from Figure 3 whether SA-T is frozen or entirely excluded from the process.

5- It’s not clear what is the benefit of the technique explained on page 4: "our approach differs from conventional video frame interpolation methods, as each frame generated through interpolation replaces the corresponding input frame. In other words, every frame in the output is newly synthesized"

6- How did authors make sure there is enough correlation between text and video segment? Details of text-video pair selection are missing.

7- In Fig. 9 (b) statistics on resolution are not clear. what “99.9%” means?

8- There is no comprehensive evaluation of diversity (since the paper claimed to improve it) besides Fig. 4. Did the authors consider evaluating diversity with human evaluation?

**Questions:**

see the weakness section.

---

> ### Author Response · Authors · 2023-11-22
> **Response from the Authors (part 1/2)**
>
> **Q1. The technical novelty of the proposed method is limited as it is a combination of existing techniques, including text-to-image generation, frame interpolation, and super-resolution.**
>
> **A1.** Since the "base-interpolation-SR" framework has become a standard design for high-quality video generation, we consider our contributions are system-level rather than model-level.
> For the entire system, we made the following contributions:
> - Model design: we propose a simple yet effective method to involve RoPE-equipped temporal self-attention in our base, interpolation and video super resolution models. Experiments show that such design is able to produce videos with temporal coherency.
> - Training: we studied different training schemes in base, interpolation and super-resolution models, and proved that when given a pre-trained text-to-image generation model, joint image-video fine-tuning is an effective way to overcome catatrophical forgetting, as well as to obtain high quality videos.
> - Evaluation: T2V is a novel video generation task, where we think novel evaluation is required. Different from other approaches by simply reporting human preference and FVD, we conduct a systematic human evaluation on various dimensions to compare with state-of-the-art.
> - Dataset: We propose a novel text-video dataset Vimeo25M to improve the performance of text-to-video generation tasks.
>
> **Q2. Dandi et al.'s work on “Jointly Trained Image and Video Generation using Residual Vectors" is relevant to this paper's exploration of joint image-video training. However, this paper is neither cited nor discussed.**
>
> **A2.**  We have included this paper in Related Works in the revised manuscript.
>
> This paper proposed a GAN-based framework to generate videos. It combined an LSTM and an image GAN to learn disentangled temporal and spatial information. Joint image-video training is conducted to learn the entire framework.
>
> Our framework is a large-scale diffusion-based system. Since diffusion model does not have a compressed latent space, our temporal module is added after each spatial block. Our joint image-video fine-tuning uses different large-scale image and video datasets, and in each batch, we concatenate images with videos in the temporal axis. Dandi et al. proposed using the same video to optimize both spatial and temporal modules. We train spatial and temporal modules jointly while Dandi et al. proposed a two-step training scheme to train the framework separately.
>
> **Q3. There is no analysis of the contribution of individual components or techniques on the video generation performance. The paper mentions the temporal module, joint image-video training, and usage of Vimeo25M dataset for training. However, we don’t see a comprehensive analysis on the impact of each of them on the performance. It’s hard to understand which component has a higher impact on the final model. The only analysis we see is in Fig 10 which is very limited by showing only three images.**
>
> **A3.**  **For the temporal module**, we have conducted small-scale experiments on UCF101 for comparison. We explore three different settings. 1)  Our proposed temporal module, 2) replacing RoPE in our module with absolute PE, 3) spatial-temporal self-attention used in [1], We train three models for same iterations and report FVD in Table
> | Method | Ours | absolute PE | spatio-temporal self-attention |
> | ---- | ----  | ---- | ---- |
> | FVD |  611 | 625 | 635 |
>
> Quantitative results demonstrate the effectiveness of our proposed temporal module.
>
> **For joint image-video tune-tuning**, we qualitatively show several examples in the updated Supplementary Material website. From the results, we can observe that without joint image-video fine-tuning, the network catastrophically forgets previous learned concepts and overfits to training data very quickly. On the opposite, using images with videos together to fine-tune the entire framework helps combine different concepts in both image and video data, which we found can help improve the diversity of the results.
>
> **For the usage of Vimeo25M data**, we have updated Tab.1 in the revised manuscript for comparison.  Without using Vimeo, we obtain higher FVD on zero-shot generation on UCF101. In addition, as shown in Tab. 4, since Vimeo25M has better aesthetics, and large-amount human data, the generated results have smoother motion and better face/body/hand quality compared with other method.
>
> **Q4. The role of the temporal self-attention module (SA-T) during image-only training phases is ambiguous. It's unclear from Figure 3 whether SA-T is frozen or entirely excluded from the process.**
>
> **A4.** Image and video are simultaneously used to train the model. Video data is used to optimize both spatial and temporal modules while images are only used to optimize the spatial part.  Since SA-T is not able to be applied to images, when optimizing the network using images, SA-T is entirely excluded from the process.

---

> > ### Author Response · Authors · 2023-11-22
> > **Response from the Authors (part 2/2)**
> >
> > **Q5. It’s not clear what is the benefit of the technique explained on page 4: "our approach differs from conventional video frame interpolation methods, as each frame generated through interpolation replaces the corresponding input frame. In other words, every frame in the output is newly synthesized"**
> >
> > **A5.** In our interpolation process, input frames are also reconstructed. In conventional video interpolation methods, input frames are directly combined with the generated frames. We found, directly reusing the input frames in generated videos leads to flickering in the final results. Generating the entire video, including input frames, makes the results have more temporally coherency.
> >
> > **Q6. How did authors make sure there is enough correlation between text and video segment? Details of text-video pair selection are missing.**
> >
> > **A6.** In the current version, we didn't conduct text-video selection process. We use the entire WebVid10M dataset, as well as filtered Vimeo25M dataset. In future work, we plan to use [2] to compute text-video correlation score for selecting videos which have higher correspodance with text.
> >
> > **Q7. In Fig. 9 (b) statistics on resolution are not clear. what “99.9%” means?**
> >
> > **A7.**  It means that the height of 99.9% of videos in the WebVid10M dataset is in the range of 400~600. We will improve the figure in the final version.
> >
> > **Q8. There is no comprehensive evaluation of diversity (since the paper claimed to improve it) besides Fig. 4. Did the authors consider evaluating diversity with human evaluation?**
> >
> > **A8.** At current stage, we use FVD to report both generated video quality and diversity, and we show improvement compared to other approaches. In future work, we plan to designed around 20 prompts and generate 10 videos per prompt using different methods. Human raters will be asked to compare which method is able to generate more diverse results per prompt. Final diversity will be reported based on the average of all the prompt results.
> >
> > [1] Wu et al, Tune-a-video: One-shot tuning of image diffusion models for text-to-video generation, CVPR 2023
> >
> > [2] Li et al., Unmasked teacher: Towards training-efficient video foundation models. ICCV 2023

---

### Author Response · Authors · 2023-11-22
**To all reviewers**

We really appreciate all four reviewers for their careful reviews and valuable comments. We have updated Supplementary Material towards better answering reviewers' questions and demonstrating our results. We add:
- Qualitative comparison on results trained with and without joint image-video fine-tuning
- Generated results beyond training length

We have uploaded a revised manuscript incorporating reviewers' feedback. Below is a summary of the main changes:
- update related works
- update Tab. 1 with FVD without using Vimeo25M
- update Fig. 1 and corresponding caption

---

### Meta-Review · Area_Chair_GUgn · 2023-12-11

**Metareview:**

The paper presents a text-to-video generation method, including text-2-video, temporal interpolation and super-resolution. The main technical contributions include temporal self-attentions, joint image-video fine-tuning, and a Vimeo25 dataset.

Overall the reviews are borderline with two leaning to accept and two leaning to reject.

The main concerns were:
- no analysis of the contribution of individual components
- technical novelty is limited, given many of the recent methods.

While not taking AnimeDiff and MagicEdit into account, the use of temporal self-attentions and joint image-video fine-tuning are indeed well-known techniques (used in the first video diffusion model by J. Ho). The AC agrees with the reviewers 9GLX and aq4u that the technical novelty is limited. The dataset contribution is interesting, but the AC believes that it itself may not be sufficient for the publication.

Considering all these issues, the AC decided to reject.

**Justification For Why Not Higher Score:**

The technical novelty in the proposed method is limited.

**Justification For Why Not Lower Score:**

N/A

---

### Decision · Program_Chairs · 2024-01-16

Reject